# Hepatotoxicity risk factors in HIV-infected MSM with HBV/HCV coinfections: A cohort study in Northwestern China

Xinyu Ma[1,2,☯], Ruimin Bai[1☯], Zirong Zhu[2,3], Tongquan Wang[4], Xinrong Zhao[1], Yan Sun[1], Yan Zheng[1*]

1 Department of Dermatology, The First Affiliated Hospital of Xi'an Jiaotong University, Xi'an Jiaotong University, Xi'an, Shaanxi Province, China, 2 Department of Dermatology, Xi'an Liver Disease Hospital (Xi'an Eighth Hospital), Xi'an, Shaanxi Province, China, 3 Department of Dermatology, Xi'an People's Hospital (Xi'an Fourth Hospital), Xi'an, Shaanxi Province, China, 4 College of Electrical Engineering, Xi'an Jiaotong University, Xi'an, Shaanxi Province, China

☯ These authors contributed equally to this work.
* zenyan66@126.com

## Abstract

### Objective

To evaluate the prevalence of hepatitis B virus (HBV) and hepatitis C virus (HCV) coinfections among human immunodeficiency virus (HIV)-infected men who have sex with men (MSM) initiating their first antiretroviral therapy (ART) in Northwestern China, and to examine the impact of coinfections on hepatotoxicity risk.

### Methods

This retrospective cohort study analyzed MSM who were newly diagnosed with HIV and initiated ART in Northwestern China between January 1st, 2005, and June 30th, 2019. A total of 4,690 MSM aged ≥18 years were included and categorized into three groups based on HBV or HCV coinfection status: HIV monoinfection, HIV/HBV coinfection, and HIV/HCV coinfection. Hepatotoxicity was classified into grades 0 (normal) to 4 (life-threatening) according to the degree of elevation in liver enzymes (AST or ALT). Kaplan-Meier curves were used to evaluate the incidence of hepatotoxicity, mortality, and ART failure rates, while Cox proportional hazards models assessed the independent impact of coinfections on hepatotoxicity risk.

### Results

Among 4,690 HIV-infected MSM, the prevalence of HIV/HBV and HIV/HCV coinfections was 5.18% and 2.17%, respectively. Coinfected individuals had significantly elevated hepatotoxicity risks. HIV/HBV coinfection substantially increased the risk of any-grade hepatotoxicity (Hazard Ratio [HR]: 1.190, 95% Confidence Interval [CI]: 1.029–1.375) and grade ≥3 hepatotoxicity (HR: 3.161, 95% CI: 2.095–4.769). HIV/

**Data availability statement:** The data underlying this study were obtained from the electronic medical record system of Xi'an Eighth Hospital and contain potentially identifying and sensitive patient information, including demographic and clinical follow-up data. Due to ethical and legal restrictions imposed by the Medical Ethics Committee of Xi'an Eighth Hospital, these data cannot be shared publicly. Researchers who meet the criteria for access to confidential data may request access by contacting the Medical Ethics Committee of Xi'an Eighth Hospital at bygcpll2018@163.com.

**Funding:** This study was financially supported by the National Natural Science Foundation of China in the form of a grant (82273541) received by YZ. No additional external funding was received for this study. The funder had no role in the study design, data collection and analysis, decision to publish, or preparation of the manuscript.

**Competing interests:** The authors have declared that no competing interests exist.

HCV coinfection was also associated with a higher risk of any-grade hepatotoxicity (HR: 1.311, 95% CI: 1.050–1.636). Older age at ART initiation, a shorter diagnosis-to-treatment interval, and HBV/HCV coinfection were identified as risk factors, while higher CD4 + T lymphocyte counts and lower hemoglobin levels were protective factors.

## Conclusion

HIV/HBV and HIV/HCV coinfections significantly increased the risk of hepatotoxicity in HIV-infected MSM receiving ART. These findings underscore the importance of vigilant liver function monitoring in coinfected patients on ART, particularly in consideration of baseline factors such as age, time to treatment, CD4 + T-cell count, and hemoglobin level, to minimize interruptions and optimize outcomes.

## Background

The HIV epidemic has become one of the most significant global public health challenges, affecting approximately 40.8 million people worldwide as of 2024 [1]. As one of the most populous countries, China faces a significant HIV/AIDS epidemic as well. Since the first case was identified in the mid-1980s, HIV has spread extensively across mainland China [2]. By 2023, an estimated 1.3 million individuals in China were living with HIV [3].

MSM represent a high-risk group for HIV transmission, with an infection rate 4.94 times higher than that of the general male population [4]. In China, MSM have shown the most significant increase in HIV transmission, rising from 2.5% of newly reported cases in 2006 to 26% in 2014 [5]. Additionally, due to their behavioral factors, MSM are at an elevated risk of viral hepatitis, particularly infections with HBV and HCV [6,7].

HIV, HBV, and HCV share common transmission routes, including the exchange of infected bodily fluids, shared needles, and blood transfusions [7]. As a result, the prevalence of HIV coinfections with HBV and HCV has increased in recent years [8]. By 2014, there were approximately 2.6 million individuals worldwide living with HIV/HBV coinfection and 2.75 million with HIV/HCV coinfection [9]. These coinfections not only exacerbate liver damage but also significantly accelerate the progression of AIDS and present new challenges for treatment [6]. Studies have shown that HIV/HBV coinfection increases the risk of liver-related mortality and is associated with more severe HIV-related immunosuppression [10,11]. Compared to HIV monoinfection, HIV/HCV coinfection is associated with markedly reduced survival rates and accelerated progression of liver fibrosis [12,13].

Since the implementation of the "Four Frees and One Care" policy in 2003, significant progress has been made in the accessibility of ART in China [5]. The updated 2024 Chinese Guidelines for Diagnosis and Treatment of HIV/AIDS recommend early initiation of ART for all individuals living with HIV, regardless of disease stage or CD4 + T-cell count [14]. While ART has substantially improved patient outcomes,

hepatotoxicity remains a major concern. This risk is particularly pronounced in individuals coinfected with HBV or HCV, whose prevalence continues to rise among MSM in China. However, few studies have systematically investigated ART-related hepatotoxicity and its risk factors in this key population, and data from Northwestern China are especially scarce.

Therefore, this study aimed to examine the prevalence of HBV and HCV coinfections among HIV-infected MSM in Northwestern China and to assess their impact on ART-related hepatotoxicity. The findings provide valuable insights for early detection and management of hepatotoxicity, helping to optimize treatment strategies, reduce severe liver toxicity, and improve patient outcomes, particularly in this understudied region.

## Methods

### Study design and participants

This retrospective cohort study included HIV-infected MSM who initiated free first-line ART at Xi'an Eighth Hospital between January 1st, 2005, and June 30th, 2019. Data were collected from the initiation of ART until the end of the study (June 30, 2019) or until death, loss to follow-up, or treatment discontinuation, whichever occurred first. All patients received a standard first-line regimen consisting of efavirenz, tenofovir disoproxil fumarate, and lamivudine.

The inclusion criteria were as follows: (1) age ≥ 18 years; (2) a confirmed AIDS diagnosis date; (3) availability of baseline laboratory results for aspartate aminotransferase (AST) and alanine aminotransferase (ALT) within three months after initiating ART; and (4) confirmed results for hepatitis B surface antigen (HBsAg) and hepatitis C antibody (anti-HCV) at the start of ART. Patients with triple infections of HIV, HBV, and HCV were excluded due to the small number of such cases (n=6). Specifically, individuals with both positive HBsAg and anti-HCV results were excluded. Based on these criteria, a total of 4,690 HIV-infected MSM were included in the study and categorized into three groups according to their coinfection status: HIV monoinfection (n=4,345), HIV/HBV coinfection (n=243), and HIV/HCV coinfection (n=102).

### Data sources and collection

Local healthcare professionals collected a comprehensive range of demographic and clinical information for each patient throughout the study period, along with follow-up observations. Demographic data included date of birth, marital status, height, and weight. Clinical characteristics included the date of HIV diagnosis, the date of ART initiation, the interval between HIV diagnosis and ART initiation, and the World Health Organization (WHO) clinical stage. The comorbidities considered in this study included the following: (1) pulmonary tuberculosis, (2) extrapulmonary tuberculosis, (3) Pneumocystis pneumonia, (4) recurrent severe bacterial pneumonia, (5) recurrent severe bacterial infections (excluding pneumonia), (6) disseminated non-tuberculous mycobacterial infection, (7) disseminated fungal infection, (8) persistent or intermittent fever (lasting more than 1 month), (9) chronic diarrhea (lasting more than 1 month), (10) oral thrush, (11) herpes zoster, (12) chronic herpes simplex, (13) Kaposi's sarcoma, (14) oral hairy leukoplakia, (15) other skin lesions, (16) esophageal candidiasis, (17) extrapulmonary cryptococcosis, (18) cytomegalovirus infection, (19) central nervous system toxoplasmosis, (20) cerebral lymphoma, (21) B-cell non-Hodgkin lymphoma, and (22) other opportunistic infections or tumors.

Laboratory tests included CD4+ and CD8+T lymphocyte counts, HIV RNA viral load, white blood cell (WBC) count, platelet count, hemoglobin levels, serum total bilirubin (T.BIL), serum creatinine (Scr), ALT, AST, random blood glucose, and baseline laboratory results for serum hepatitis B surface antigen (HBsAg) and hepatitis C virus antibody (anti-HCV) at the time of ART initiation. HIV antibody positivity was initially detected using an enzyme-linked immunosorbent assay (ELISA) and confirmed by Western blot. HBsAg positivity and anti-HCV antibody positivity were determined using ELISA.

After the initiation of ART, viral load, CD4+T cell counts, and liver function test results were extracted from the ART follow-up database. These parameters were measured at baseline, at the 3rd and 6th months, and every 6 months thereafter. Data completeness was assessed for all variables, particularly follow-up laboratory results. In cases where baseline

laboratory results or follow-up laboratory data were missing, these cases were excluded from the analysis. For any missing values in variables, no imputation was performed, and the study was conducted using available case data.

ALT and AST levels were graded based on the National Cancer Institute (NCI) toxicity grading criteria: normal (upper limit of normal [ULN], 40 U/L); mild (ULN to 2.5×ULN); moderate (2.5–5×ULN); severe (5–20×ULN); and life-threatening (>20×ULN). Hepatotoxicity grades were determined using the higher value of ALT or AST.

ART failure was defined as virological failure, immunological failure, or death. Immunological failure was defined as a CD4+T lymphocyte count falling below 100 cells/mm³ or dropping to levels lower than baseline without accompanying illness after three months of ART. Virological failure was defined as a plasma viral load exceeding 1,000 copies/mL after three months of treatment, based on two consecutive measurements.

All patients were tested using standardized protocols, and data were entered into an Excel spreadsheet. Special quality control personnel proofread the data, and Excel was used for duplicate checking to ensure its authenticity and reliability.

## Statistical analysis

Data were processed using Excel 2016 (Microsoft, Redmond, WA, USA) and SPSS version 20.0 (SPSS, Inc., Chicago, IL, USA). The relationships between categorical variables were assessed using the chi-square test or Fisher's exact test, while the relationships between continuous variables were analyzed using the Kruskal-Wallis H test. A p-value of < 0.05 was considered statistically significant. For pairwise comparisons, Bonferroni correction was applied, and a p-value of < 0.0167 was considered statistically significant. Hepatotoxicity incidence and mortality rates were calculated as the number of new hepatotoxicity events and deaths per 1,000 person-years (py) of follow-up. For the incidence rate, hepatotoxicity events were counted annually, and the rate was determined by dividing the number of events by the total person-years at risk, then multiplying by 1,000. A 95% CI was chosen for the rates among the three HIV groups.

Hepatotoxicity was assessed using two endpoints: (1) the occurrence of at least one episode of hepatotoxicity of any grade, and (2) the occurrence of at least one episode of grade 3 or higher hepatotoxicity. Kaplan-Meier analysis and log-rank tests were used to compare the incidence of hepatotoxicity across the groups. Cox proportional hazards models were employed to analyze factors influencing hepatotoxicity. The proportional hazards assumption was evaluated using both graphical methods and statistical tests based on Schoenfeld residuals. All hypothesis tests were two-sided, with a p-value of <0.05 considered statistically significant.

## Ethical approval

This study was conducted by the ethical standards outlined in the Declaration of Helsinki (2013 revision) [15]. It was approved by the Ethics Committee of Xi'an Eighth Hospital on July 6, 2020 (Approval number: 2020−30). The ethics committee agreed to waive informed consent because the study involved a retrospective analysis of anonymized data collected from hospital databases.

Following ethical approval, the data for this study were accessed on August 1, 2020. To ensure the confidentiality of participants, all data are anonymized before being made available to researchers. The author did not have access to personally identifiable information during or at any time after the data collection. This ensures that the study follows ethical guidelines and protects the privacy of participants.

## Results

### Trends in HIV coinfections

Among the 4,690 HIV-infected MSM patients, 7.35% (345/4,690) had coinfections with HBV or HCV, with 5.18% (243/4,690) being HIV/HBV coinfections and 2.17% (102/4,690) being HIV/HCV coinfections (Table 1). From 2005 to 2019, the number of HIV-infected MSM patients showed an overall increasing trend. The proportion of HIV/HBV

**Table 1. Number and proportion of HIV coinfections with HBV/HCV among MSM patients from 2005 to 2019.**

| Year | Total Number | HIV monoinfection (%) | HIV/HBV coinfection (%) | HIV/HCV coinfection (%) | Coinfections[a] (%) |
|---|---|---|---|---|---|
| 2005 | 2 | 2 (100) | 0 (0) | 0 (0) | 0 (0) |
| 2006 | 2 | 2 (100) | 0 (0) | 0 (0) | 0 (0) |
| 2007 | 3 | 1 (33.33) | 1 (33.33) | 1 (33.33) | 2 (66.67) |
| 2008 | 6 | 5 (83.33) | 1 (16.67) | 0 (0) | 1 (16.67) |
| 2009 | 13 | 13 (100) | 0 (0) | 0 (0) | 0 (0) |
| 2010 | 36 | 28 (77.78) | 5 (13.89) | 3 (8.33) | 8 (22.22) |
| 2011 | 68 | 59 (86.76) | 8 (11.76) | 1 (1.47) | 9 (13.23) |
| 2012 | 178 | 168 (94.38) | 9 (5.06) | 1 (0.56) | 10 (5.62) |
| 2013 | 487 | 450 (92.40) | 27 (5.54) | 10 (2.05) | 37 (7.59) |
| 2014 | 576 | 525 (91.15) | 40 (6.94) | 11 (1.91) | 51 (8.85) |
| 2015 | 718 | 672 (93.59) | 34 (4.74) | 12 (1.67) | 46 (6.41) |
| 2016 | 741 | 681 (91.90) | 40 (5.40) | 20 (2.70) | 60 (8.10) |
| 2017 | 787 | 723 (91.87) | 41 (5.21) | 23 (2.92) | 64 (8.13) |
| 2018 | 689 | 649 (94.19) | 24 (3.48) | 16 (2.32) | 40 (5.80) |
| 2019[b] | 384 | 367 (95.57) | 13 (3.39) | 4 (1.04) | 17 (4.43) |
| **Total** | 4690 | 4345 (92.64) | 243 (5.18) | 102 (2.17) | 345 (7.35) |

[a]Coinfections include HIV/HBV coinfection and HIV/HCV coinfection;

[b]For 2019, the number of HIV-infected MSM drops because only half a year's data is available.

MSM, men who have sex with men.

coinfections also demonstrated an upward trend (Fig 1). In contrast, the proportion of HIV/HCV coinfections remained low with minimal fluctuations over the same period.

## Baseline characteristics

As shown in Table 2, among the 4,690 HIV-infected MSM patients, the majority (63.5%) were aged 30–49 years, with a median age of 39 years (IQR: 33–49 years). Patients with HIV/HBV coinfection were significantly older than those with HIV monoinfection (P = 0.004). Across all three groups, the proportion of unmarried individuals was high. Additionally, patients with HIV/HBV coinfection started ART at an older age compared to those with HIV monoinfection.

Compared to HIV monoinfection, patients with HIV/HCV coinfection had a longer interval between HIV diagnosis and ART initiation. HIV/HBV coinfected patients had lower CD4 lymphocyte counts, platelet counts, and white blood cell counts, as well as higher AST and ALT levels, compared to those with HIV monoinfection. The distribution of hepatotoxicity grades also differed significantly between these two groups.

No significant differences were observed among the three groups in BMI, current WHO clinical stage, CD8 lymphocyte counts, HIV viral load, total bilirubin, serum creatinine, or the presence of comorbidities. After Bonferroni correction, all of the above findings remained statistically significant (P < 0.05), as shown in Fig 2.

## Hepatotoxicity incidence and influencing factors

We followed 4,690 HIV-infected MSM patients, of whom 3,562 developed hepatotoxicity, with an incidence rate of 855.6 per 1,000 person-years. Additionally, 194 patients experienced grade ≥3 hepatotoxicity, corresponding to an incidence rate of 14.6 per 1,000 person-years (Table 3). For hepatotoxicity of any grade, HIV/HCV coinfected patients had the highest incidence rate, followed by those with HIV/HBV coinfection. In contrast, patients with HIV monoinfection had the lowest incidence (all P < 0.001). However, for grade ≥3 hepatotoxicities, patients with HIV/HBV coinfection had a higher incidence rate than those with HIV/HCV coinfection, and patients with HIV monoinfection had the lowest rate (all P < 0.001).

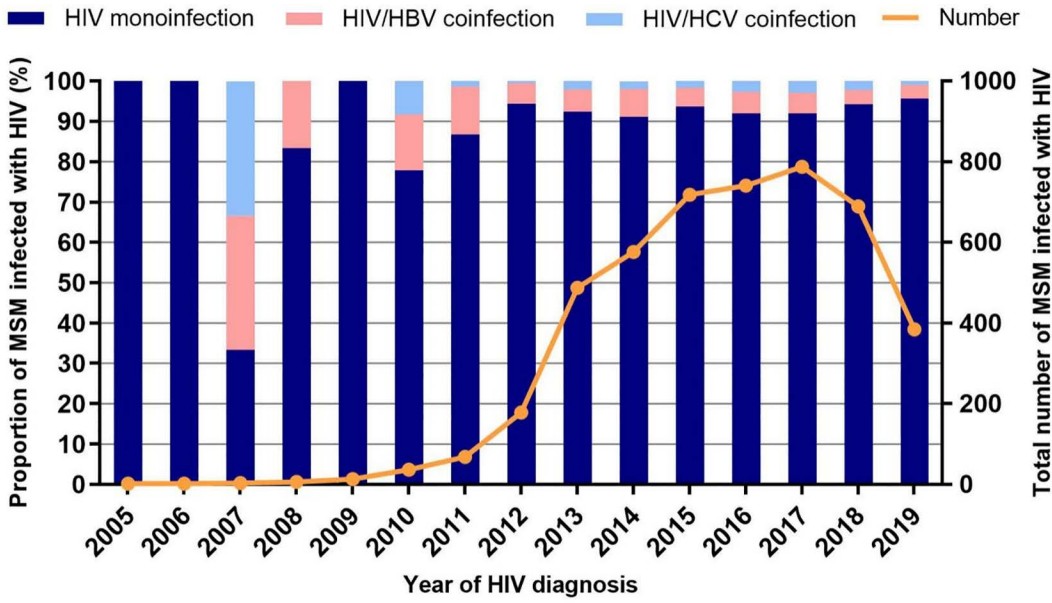

**Fig 1. Dynamic trends in the number and proportion of MSM patients with HIV coinfections of HBV/HCV from 2005 to 2019.** For 2019, the number of HIV-infected MSM drops because only half a year's data is available. MSM, men who have sex with men.

Kaplan-Meier survival analysis and log-rank tests revealed significant differences in the incidence rates of hepatotoxicity of any grade and grade ≥3 hepatotoxicity among the three groups (all P < 0.001). The HIV/HCV coinfection group exhibited a rapid increase in any-grade hepatotoxicity during the early stages of follow-up, but the growth rate of severe hepatotoxicity slowed in later stages. In contrast, the HIV/HBV coinfection group demonstrated a more sustained risk of severe hepatotoxicity over time, as shown in Fig 3.

Univariate analysis identified eight factors associated with hepatotoxicity, including coinfection status, age, marital status, age at ART initiation, the interval between HIV diagnosis and ART initiation, CD4 + lymphocyte count, platelet count, and hemoglobin level (all P < 0.1). Cox proportional hazards further revealed that HIV/HBV and HIV/HCV coinfections were among the key factors influencing the occurrence of hepatotoxicity in HIV-infected MSM patients.

Compared to MSM with HIV monoinfection, those with HIV/HBV coinfection (HR: 1.190, 95% CI: 1.029–1.375) and HIV/HCV coinfection (HR: 1.311, 95% CI: 1.050–1.636) had a significantly higher risk of developing hepatotoxicity of any grade (Table 4). For grade ≥3 hepatotoxicities, MSM with HIV/HBV coinfection had a markedly increased risk (HR: 3.161, 95% CI: 2.095–4.769) compared to those with HIV monoinfection, whereas the risk among HIV/HCV coinfected patients did not differ significantly from that of the HIV monoinfection group.

The Cox proportional hazards model also demonstrated that older age at ART initiation, a shorter diagnosis-to-treatment interval, and HBV/HCV coinfection were significant risk factors for hepatotoxicity of any grade. In contrast, higher CD4 + lymphocyte counts and lower hemoglobin levels were identified as protective factors against hepatotoxicity of any grade. Table 4 summarizes the multivariate HR for the major predictors of hepatotoxicity and their 95% CI. Fig 4 summarizes the risk and protective factors for ART-related hepatotoxicity.

During the follow-up period, a total of 557 treatment regimen changes were recorded, with 267 cases (47.9%) attributed to adverse effects. Among these adverse effects, liver function abnormalities were the most common (81 cases, 30.3%), followed by bone marrow suppression (45 cases, 18.4%).

**Table 2. Comparison of baseline characteristics among the three infection status groups in the MSM population.**

| Characteristic | ALL | HIV monoinfection | HIV/HBV coinfection | HIV/HCV coinfection | P | Pa | Pb | Pc |
|---|---|---|---|---|---|---|---|---|
| Number | 4690 | 4345 | 243 | 102 | | | | |
| **Age (years)*** | | | | | **0.002** | **0.004** | 0.233 | 1.000 |
| Years, median (IQR) | 39 (33–49) | 39 (33–49) | 42 (35–51) | 41 (33–54) | | | | |
| **Marital status [number (%)]*** | | | | | **0.048** | 0.055 | 0.172 | 0.516 |
| Unmarried | 2723 (58.1) | 2545 (58.6) | 124 (51.0) | 54 (52.9) | | | | |
| Married/live together | 1341 (28.6) | 1236 (28.4) | 75 (30.9) | 30 (29.4) | | | | |
| Divorce/Separation | 596 (12.7) | 537 (12.4) | 43 (17.7) | 16 (15.7) | | | | |
| Loss of spouse | 29 (0.62) | 26 (0.6) | 1 (0.4) | 2 (2.0) | | | | |
| **BMI (kg/m²)** | | | | | 0.991 | – | – | – |
| Median (IQR) | 21.5 (19.6–23.5) | 21.5 (19.6–23.6) | 21.7 (20.0–23.0) | 21.3 (19.8–23.4) | | | | |
| **Age at ART initiation*** | | | | | **0.008** | **0.030** | 0.207 | 1.000 |
| Years, median (IQR) | 31 (26–42) | 31 (25–41) | 34 (28–42) | 34 (26–47) | | | | |
| **The interval between diagnosis and treatment*** | | | | | **0.007** | 1.000 | **0.007** | 0.103 |
| Days, median (IQR) | 33 (17–103) | 33 (16–99) | 34 (18–138) | 53 (22–220) | | | | |
| **WHO clinical stage [number (%)]** | | | | | 0.291 | – | – | – |
| Stage I | 2280 | 2119 (49.0) | 110 (45.5) | 51 (50.0) | | | | |
| Stage II | 914 | 846 (19.6) | 51 (21.1) | 17 (16.8) | | | | |
| Stage III | 787 | 737 (17.0) | 38 (15.7) | 12 (11.9) | | | | |
| Stage IV | 689 | 625 (14.4) | 43 (17.8) | 21 (20.8) | | | | |
| **CD4+lymphocyte counts (/µL)*** | | | | | **0.008** | **0.007** | 0.783 | 1.000 |
| Median (IQR) | 309 (184–441) | 312 (188–443) | 283 (125–388) | 288.5 (177.5–423.5) | | | | |
| **CD8+lymphocyte counts (/µL)** | | | | | 0.142 | – | – | – |
| Median (IQR) | 1058 (727–1512) | 1034 (715–1465) | 1036.5 (748.0–1376.3) | 868.0 (504.5–1577.5) | | | | |
| **HIV RNA load (copies/ml)** | | | | | 0.689 | – | – | – |
| Median (IQR) | 40114 (12375–163500) | 40686.5 (12125.0–169181.3) | 44500 (18000–93175) | 21000.0 (6278.0–179800) | | | | |
| **White blood cell counts ($10^9$/L)*** | | | | | **<0.001** | **<0.001** | 1.000 | 0.063 |
| Median (IQR) | 5.1 (4.2–6.1) | 5.1 (4.2–6.2) | 4.7 (3.8–5.8) | 5.2 (4.1–6.1) | | | | |
| **Platelet counts ($10^9$/L)*** | | | | | **<0.001** | **<0.001** | 0.165 | 0.582 |
| Median (IQR) | 183 (149–220) | 184 (151–221) | 162.5 (126.3–204.8) | 175.5 (139.5–206.8) | | | | |
| **Hemoglobin (g/L)*** | | | | | **0.038** | 0.161 | 0.240 | 1.000 |
| Median (IQR) | 149 (138–157) | 149 (139–157) | 147.5 (130.3–157.8) | 146.0 (135.5–155.0) | | | | |
| **AST (U/L)*** | | | | | **<0.001** | **<0.001** | 1.000 | 0.012 |
| Median (IQR) | 22.9 (19.0–28.7) | 22.6 (19.0–28.2) | 26 (20.0–34.9) | 23.0 (19.3–28.0) | | | | |
| **ALT (U/L)*** | | | | | **<0.001** | **<0.001** | 0.360 | 0.782 |
| Median (IQR) | 22.3 (15.8–33.9) | 22.0 (15.6–33.3) | 26.0 (18.0–43.8) | 24.0 (17.4–33.5) | | | | |
| **Hepatotoxicity classification [number (%)]*** | | | | | **<0.001** | **<0.001** | 1.000 | 0.083 |
| Grade 0 | 3773 (80.4) | 3526 (81.2) | 164 (67.5) | 83 (81.4) | | | | |
| Grade 1 | 861 (18.4) | 771 (17.7) | 72 (29.6) | 18 (17.6) | | | | |
| Grade 2 | 50 (1.1) | 45 (1.0) | 4 (1.6) | 1 (1.0) | | | | |
| Grade 3 | 5 (0.1) | 3 (0.1) | 2 (0.8) | 0 (0.0) | | | | |
| Grade 4 | 1 (0.0) | 0 (0.0) | 1 (0.4) | 0 (0.0) | | | | |

*(Continued)*

**Table 2.** (Continued)

| Characteristic | ALL | HIV monoinfection | HIV/HBV coinfection | HIV/HCV coinfection | P | Pa | Pb | Pc |
|---|---|---|---|---|---|---|---|---|
| **Serum total bilirubin (μmol/L)** | | | | | 0.933 | – | – | – |
| Median (IQR) | 12.1 (9.1–15.7) | 12.1 (9.1–15.7) | 12.8 (9.4–16.6) | 13.1 (9.7–17.2) | | | | |
| **Serum creatinine (μmol/L)** | | | | | 0.535 | – | – | – |
| Median (IQR) | 69.6 (63.2–76.9) | 69.6 (63.3–76.9) | 69.6 (63.0–77.7) | 68.8 (61.0–76.2) | | | | |
| **Comorbidities [number (%)]** | | | | | 0.335 | – | – | – |
| With Comorbidities | 947 (20.2) | 876 (20.2) | 55 (22.6) | 16 (15.7) | | | | |
| Without Comorbidities | 3743 (79.8) | 3469 (79.8) | 188 (77.4) | 86 (84.3) | | | | |

P values are calculated for comparisons among HIV monoinfection, HIV/HBV coinfection, and HIV/HCV coinfection;

aPa measured the difference between HIV monoinfection and HIV/HBV coinfection;

bPb measured the difference between HIV monoinfection and HIV/HCV coinfection;

cPc measured the difference between HIV/HBV coinfection and HIV/HCV coinfection;

* Characteristics with statistically significant differences among the three groups are selected for further analysis, which are indicated using a * in the superscript.

MSM, men who have sex with men; ART, antiretroviral therapy; AST, aspartate aminotransferase; ALT, alanine aminotransferase.

Additionally, 399 cases of treatment discontinuation were observed, with poor adherence being the most frequent cause (360 cases, 90.2%). Adverse effects accounted for 18 cases of treatment discontinuation, with liver function abnormalities being the most common (6 cases, 33.3%).

## Mortality and response to ART

During the study period, 63 patients died, resulting in an overall mortality rate of 4.6 per 1,000 person-years. The mortality rate among MSM patients with HIV/HBV coinfection did not differ significantly from that of those with HIV monoinfection, and no deaths were observed among patients with HIV/HCV coinfection during the follow-up period.

In the HIV monoinfection group, 57 deaths occurred (1.1%), while 6 deaths (2.5%) were reported in the HIV/HBV coinfection group. The mortality rates were 4.5 per 1,000 person-years (95% CI: 3.4–5.7) for the monoinfection group and 7.4 per 1,000 person-years (95% CI: 2.5–13.6) for the HIV/HBV coinfection group. Kaplan-Meier survival curves showed no significant differences in mortality among the three groups (P = 0.920, Fig 5a).

During the follow-up period from three months after ART initiation to the end of the study, 177 patients experienced immunological failure, and 1 patient experienced virological failure, who belonged to the HIV monoinfection group. The immunological failure occurred in 3.8% (167/4,339) of patients with HIV monoinfection, 2.0% (5/248) of those with HIV/HBV coinfection, and 4.9% (5/103) of those with HIV/HCV coinfection. There were no significant differences in the likelihood of ART failure among the three groups (P = 0.672, Fig 5b).

## Discussion

This study explored the prevalence and impact of HBV and HCV coinfections on ART-related hepatotoxicity among HIV-infected MSM in Northwestern China.

This study found that HIV coinfections with HBV or HCV significantly increased the risk of hepatotoxicity during ART among MSM, consistent with findings from studies in other countries and regions [11,16–18]. In a large-scale research following the introduction of ART, Sulkowski et al. reported that HIV/HCV coinfected individuals were more likely to develop liver enzyme abnormalities compared to those with HIV monoinfection [19]. Similarly, Thio et al. highlighted that HIV/HBV coinfection was significantly associated with severe hepatotoxicity and liver decompensation [10].

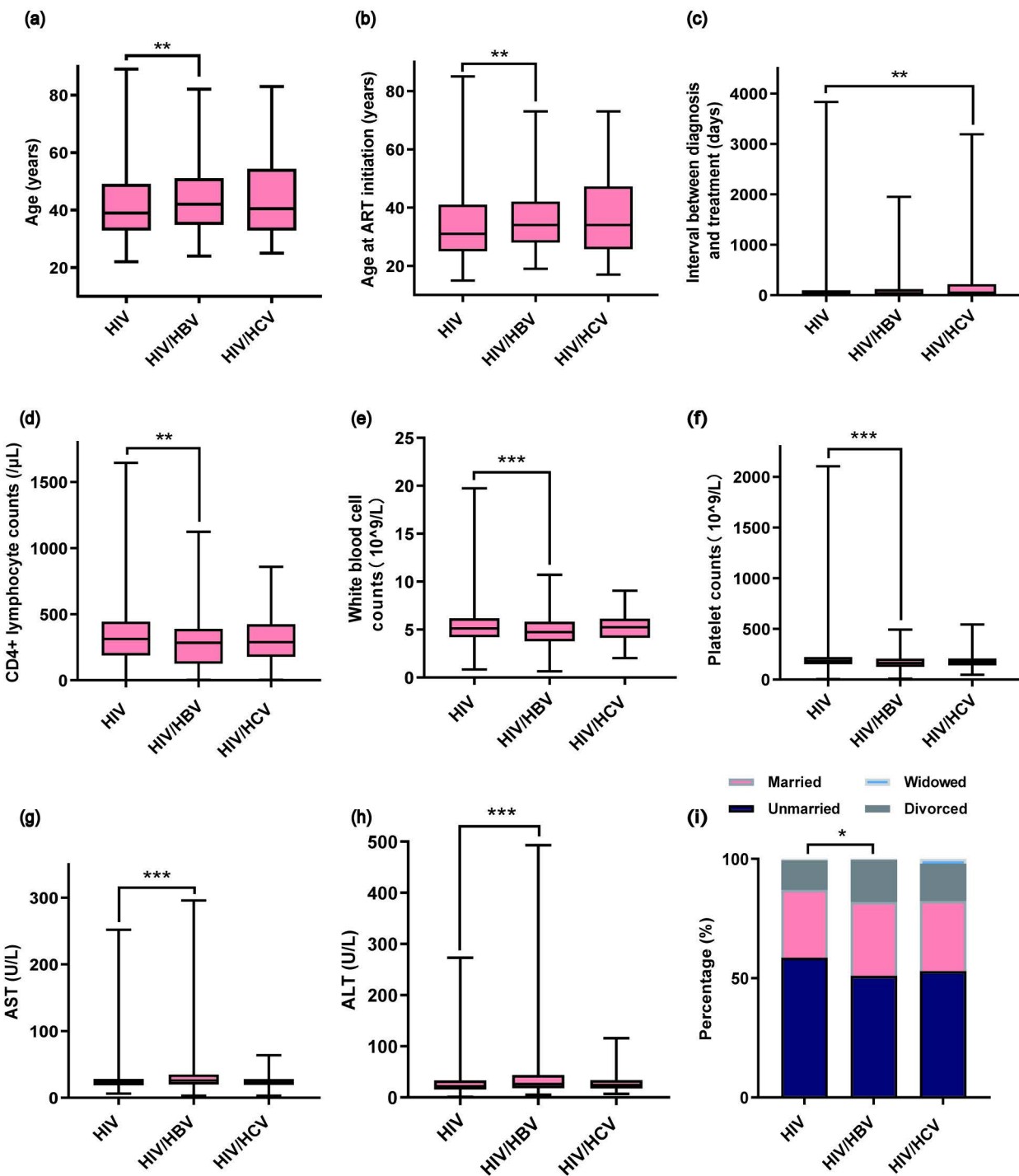

**Fig 2. Comparison of baseline characteristics of MSM grouped by infection status, including (a) age, (b) age at ART initiation, (c) interval between diagnosis and treatment, (d) CD4＋lymphocyte counts, (e) White blood cell counts, (f) platelet counts, (g) AST, (h) ALT, (i) marital status.** Statistically significant pairwise comparison results are indicated as follows: a single asterisk (*) for 0.01 < P < 0.05, double asterisks (**) for 0.001 < P < 0.01, and triple asterisks (***) for P < 0.001. MSM, men who have sex with men; ART, antiretroviral therapy; AST, aspartate aminotransferase; ALT, alanine aminotransferase.

**Table 3. Standardized incidence rates of hepatotoxicity among MSM grouped by infection status.**

| Hepatotoxicity | n | Any grade | | | Grade ≥ 3 | | |
|---|---|---|---|---|---|---|---|
| | | events | py | SR (95%CI) | events | py | SR (95%CI) |
| **Total** | 4690 | 3562 | 4163.3 | 855.6 (827.5–883.7) | 194 | 13308.1 | 14.6 (12.5–16.6) |
| **HIV monoinfection** | 4345 | 3281 | 3900.5 | 841.2 (812.4–867.0) | 160 | 12302.7 | 13.0 (11.0–15.0) |
| **HIV/HBV coinfection** | 243 | 198 | 200.7 | 986.6 (849.2–1124.1) | 27 | 709.3 | 38.0 (23.7–52.4) |
| **HIV/HCV coinfection** | 102 | 83 | 62.1 | 1336.4 (1048.9–1623.9) | 7 | 296.0 | 23.7 (6.1–41.2) |

MSM, men who have sex with men; Py, person-year; SR, standardized incidence rates of hepatotoxicity; SR = events/py × 1000.

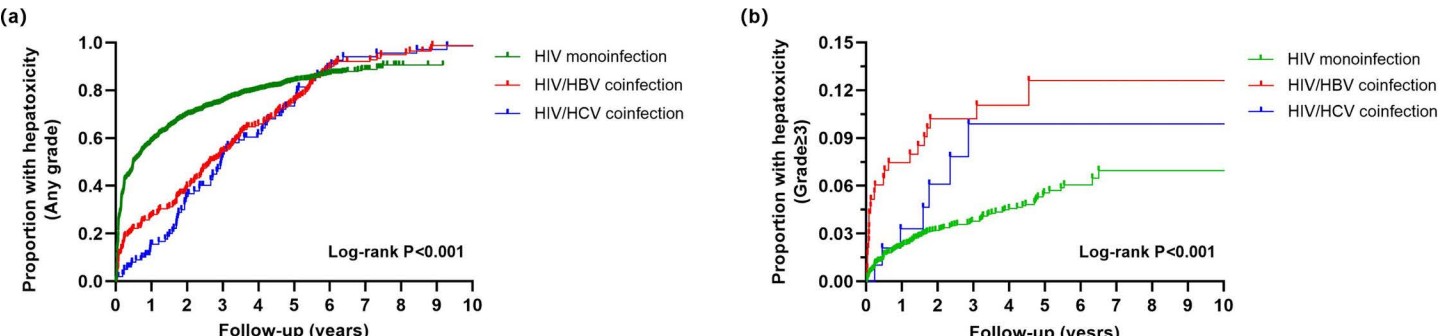

**Fig 3. Kaplan-Meier survival curves for hepatotoxicity incidence among MSM grouped by infection status.** (**a**) any grade, (**b**) grade ≥ 3. MSM, men who have sex with men.

This study confirms these trends and provides a comparative analysis of hepatotoxicity incidence rates across varying degrees of severity between MSM with HIV/HBV and HIV/HCV coinfections. It was observed that MSM with HIV/HCV coinfection had a significantly higher overall incidence of hepatotoxicity compared to those with HIV/HBV coinfection. However, MSM with HIV/HBV coinfection exhibited a greater risk of grade ≥3 hepatotoxicity. Previous studies have also suggested that HCV coinfection has a stronger association with overall hepatotoxicity compared to HBV coinfection, although the probability of severe hepatotoxicity remains low [18,20].

This phenomenon could be attributed to the direct hepatotoxic effects of HCV [21], coupled with the immunosuppressive effects of HIV, which accelerate the progression of HCV infection [22]. As a result, HIV/HCV coinfected patients tend to exhibit a higher overall incidence of hepatotoxicity. In contrast, among HIV/HBV coinfected patients, the immune escape mechanisms of HBV and the antiviral effects of treatment may reduce the overall incidence of hepatotoxicity [23]. However, after immune recovery, some patients might experience immune reconstitution inflammatory syndrome (IRIS) [24], leading to an increased risk of grade ≥3 severe hepatotoxicity. This suggests the need for more proactive liver function monitoring and early intervention for HIV/HCV coinfected individuals, particularly in the early stages of infection. For HIV/HBV coinfected individuals, attention should be focused on the long-term risk of hepatotoxicity, with targeted interventions for severe liver damage.

Regarding ART-related hepatotoxicity, previous studies have demonstrated that the metabolic pathways of different ART drugs significantly influence the incidence of hepatotoxicity [25]. In this study, since all patients received a standard first-line regimen consisting of efavirenz, tenofovir disoproxil fumarate, and lamivudine, ART drug classes were not explicitly differentiated. However, it was observed that MSM patients who initiated ART earlier faced a higher risk of hepatotoxicity, which aligns with the cumulative toxic effects associated with ART drugs. Additionally, liver function abnormalities were identified as the most common adverse effect during ART in this study, highlighting the critical role of hepatotoxicity

**Table 4. Cox proportional hazards analysis of factors associated with hepatotoxicity.**

| Hepatotoxicity | Any grade | | | | Grade ≥ 3 | | | |
|---|---|---|---|---|---|---|---|---|
| | Univariate analysis | P[a] | Multivariable analysis | P[b] | Univariate analysis | P[c] | Multivariable analysis | P[d] |
| **Infection status** | | 0.011 | | 0.004 | | <0.001 | | <0.001 |
| HIV monoinfection | 1.000 | | 1.000 | | 1.000 | | 1.000 | |
| HIV/HBV coinfection | 1.197 (1.037–1.382) | 0.014 | **1.190 (1.029–1.375)** | 0.019 | 3.025 (2.012–4.548) | <0.001 | **3.161 (2.095–4.769)** | <0.001 |
| HIV/HCV coinfection | 1.222 (0.983–1.519) | 0.072 | **1.311 (1.050–1.636)** | 0.017 | 1.834 (0.860–3.910) | 0.116 | 2.020 (0.946–4.312) | 0.069 |
| **Age (years)** | | <0.001 | | 0.335 | | 0.064 | | 0.438 |
| <33 | 1.000 | | 1.000 | | 1.000 | | 1.000 | |
| 33–38 | 1.173 (1.064–1.292) | 0.001 | 0.963 (0.841–1.103) | 0.587 | 1.487 (0.981–2.253) | 0.062 | 1.080 (0.605–1.926) | 0.795 |
| 39–48 | 1.326 (1.205–1.459) | <0.001 | 0.908 (0.752–1.096) | 0.315 | 1.121 (0.725–1.733) | 0.609 | 0.763 (0.346–1.680) | 0.502 |
| ≥49 | 1.151 (1.045–1.267) | 0.004 | 0.789 (0.608–1.023) | 0.074 | 0.920 (0.587–1.443) | 0.716 | 1.212 (0.415–3.542) | 0.725 |
| **Marital status** | | 0.004 | | 0.559 | | 0.153 | – | – |
| Unmarried | 1.000 | | 1.000 | | 1.000 | | – | – |
| Married/live together | 1.134 (1.052–1.221) | 0.001 | 1.090 (0.985–1.207) | 0.094 | 0.708 (0.506–0.990) | 0.043 | – | – |
| Divorce/Separation | 1.126 (1.017–1.247) | 0.022 | 1.069 (0.945–1.210) | 0.289 | 0.682 (0.421–1.106) | 0.121 | – | – |
| Loss of spouse | 1.016 (0.683–1.649) | 0.791 | 1.024 (0.652–1.611) | 0.919 | 1.824 (0.451–7.374) | 0.399 | – | – |
| **Age at ART initiation (years)** | | <0.001 | | <0.001 | | 0.019 | | 0.128 |
| <25 | 1.000 | | 1.000 | | 1.000 | | 1.000 | |
| 25–30 | 1.225 (1.110–1.353) | <0.001 | **1.251 (1.092–1.434)** | 0.001 | 1.588 (1.041–2.423) | 0.032 | 1.544 (0.861–2.770) | 0.145 |
| 31–40 | 1.451 (1.316–1.600) | <0.001 | **1.519 (1.256–1.837)** | <0.001 | 1.332 (0.864–2.054) | 0.195 | 1.634 (0.741–3.604) | 0.223 |
| ≥41 | 1.227 (1.112–1.354) | <0.001 | **1.429 (1.098–1.860)** | 0.009 | 0.902 (0.565–1.439) | 0.665 | 0.798 (0.265–2.397) | 0.687 |
| **Interval between diagnosis and treatment (days)** | | <0.001 | | 0.004 | | 0.622 | – | – |
| <21 | 1.173 (1.080–1.273) | <0.001 | **1.143 (1.053–1.241)** | 0.002 | 1.000 | | – | – |
| 21–57 | 1.122 (1.036–1.215) | 0.005 | **1.125 (1.036–1.223)** | 0.009 | 0.851 (0.595–1.216) | 0.397 | – | – |
| ≥58 | 1.000 | | 1.000 | | 0.996 (0.713–1.391) | 0.978 | – | – |
| **CD4 + lymphocyte counts (/µL)** | | <0.001 | | <0.001 | | 0.073 | | 0.055 |
| <188 | 1.000 | | 1.000 | | 1.000 | | 1.000 | |
| 188–311 | 0.757 (0.689–0.831) | <0.001 | **0.753 (0.682–0.832)** | <0.001 | 0.818 (0.533–1.256) | 0.358 | 0.836 (0.543–1.285) | 0.414 |
| 312–442 | 0.789 (0.719–0.865) | <0.001 | **0.791 (0.715–0.875)** | <0.001 | 0.992 (0.659–1.493) | 0.968 | 0.979 (0.648–1.480) | 0.921 |
| ≥443 | 0.838 (0.764–0.919) | <0.001 | **0.846 (0.764–0.937)** | 0.001 | 1.366 (0.928–2.010) | 0.113 | 1.421 (0.958–2.108) | 0.081 |
| **Platelet (10$^9$/L)** | | 0.026 | | 0.016 | | 0.745 | – | – |
| <162 | 1.000 | | 1.000 | | 1.000 | | – | – |
| 162–204 | 0.938 (0.864–1.017) | 0.121 | 0.963 (0.885–1.047) | 0.374 | 1.025 (0.721–1.458) | 0.887 | – | – |
| ≥205 | 1.047 (0.966–1.135) | 0.259 | 1.082 (0.995–1.177) | 0.065 | 1.134 (0.805–1.599) | 0.563 | – | – |
| **Hemoglobin (g/L)** | | 0.009 | | 0.001 | | 0.592 | – | – |
| <139 | 0.951 (0.869–1.042) | 0.284 | **0.834 (0.753–0.923)** | <0.001 | 1.000 | | – | – |
| 139–148 | 0.862 (0.786–0.945) | 0.002 | **0.841 (0.766–0.924)** | <0.001 | 0.750 (0.492–1.144) | 0.736 | – | – |
| 149–156 | 0.898 (0.820–0.984) | 0.001 | **0.890 (0.812–0.976)** | 0.013 | 0.912 (0.614–1.355) | 0.292 | – | – |
| ≥157 | 1.000 | | 1.000 | | 0.937 (0.639–1.372) | 0.892 | – | – |

[a] P[a] measured the association between factors and any grade of hepatotoxicity in univariate analysis;

[b] P[b] measured the association between factors and any grade of hepatotoxicity in multivariable analysis;

[c] P[c] measured the association between factors and grade ≥ 3 of hepatotoxicity in univariate analysis;

[d] P[d] measured the association between factors and grade ≥ 3 of hepatotoxicity in multivariable analysis.

ART, antiretroviral therapy.

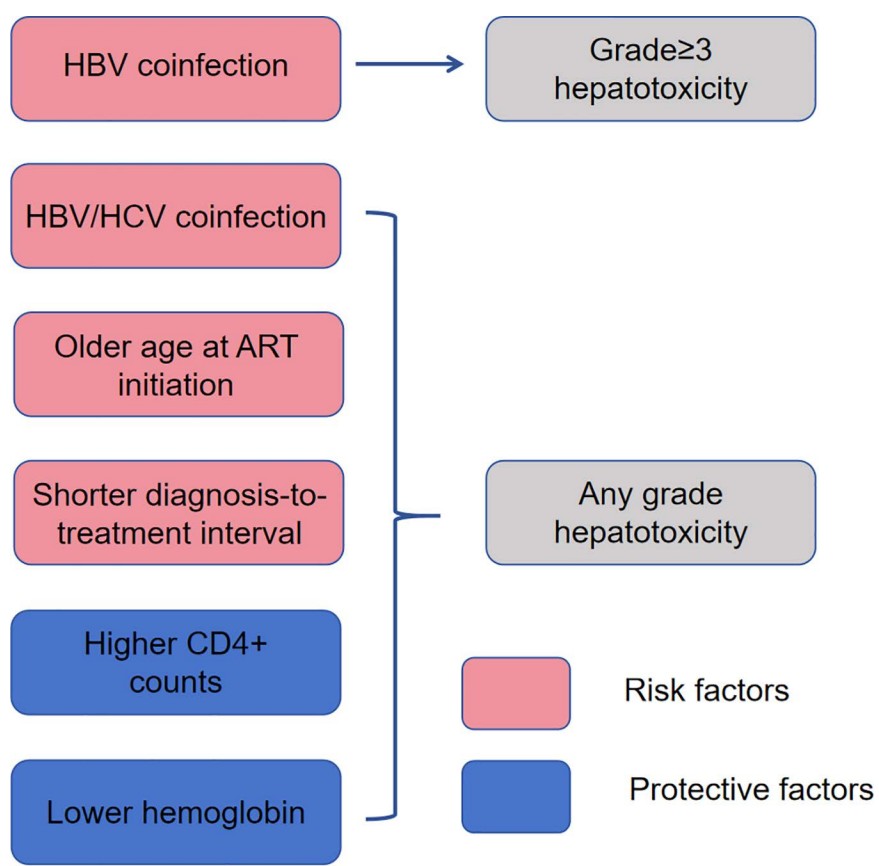

**Fig 4. Risk and protective factors for ART- related hepatotoxicity. ART, antiretroviral therapy.**

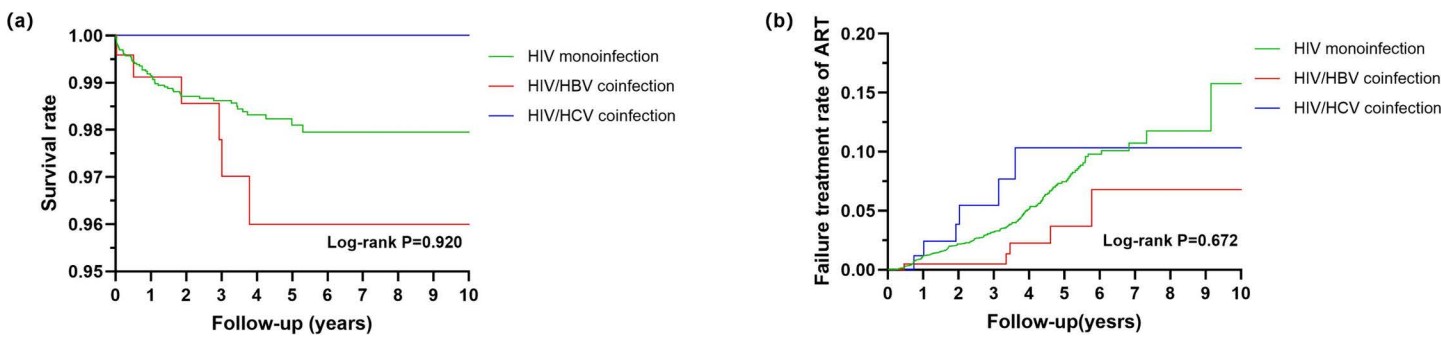

**Fig 5. Kaplan-Meier survival curves for mortality and ART failure rates among MSM grouped by infection status. (a)** survival; **(b)** ART failure. MSM, men who have sex with men; ART, antiretroviral therapy.

in influencing treatment adjustments and patient adherence. This finding underscores the importance of liver function monitoring throughout ART.

The study also found that a higher baseline CD4+T lymphocyte count had a protective effect against hepatotoxicity, consistent with previous findings [26,27]. This may be explained by the fact that a better immune status reduces direct

hepatic damage caused by viral replication while enhancing the body's resistance to inflammation and fibrosis progression. This study further highlights the important role of immune status in regulating liver complications during ART.

In addition, this study found no significant difference in mortality rates between MSM patients with HIV/HBV coinfection and those with HIV monoinfection. At the same time, no deaths were observed among MSM patients with HIV/HCV coinfection. There was no significant difference in immunological failure rates between coinfected and monoinfected patients. Shuo Feng's study in Northwestern China also reported no significant differences in mortality rates between coinfected and monoinfected patients [28]. This finding aligns with the results of this study due to the shared geographic region. A survey by Jules B. Tchatchueng Mbougua in Cameroon demonstrated similar results, showing no significant differences in mortality or antiretroviral therapy response between patients with HIV/HBV or HIV/HCV coinfections and those with HIV monoinfection [29].

In contrast, a study by Johan van Griensven conducted in Cambodia and a multicenter cohort study of AIDS patients in the Asia-Pacific region both reported that MSM with HIV/HCV coinfection had higher mortality rates than those with HIV monoinfection, while ART responses showed no significant differences. These studies also found that mortality rates for HIV/HBV coinfected individuals were comparable to those of monoinfected patients, but HIV/HBV coinfected patients exhibited lower CD4 recovery rates after ART [30,31]. Similarly, a study by Jingya Jia in southwestern China reported significantly higher mortality rates in patients with HIV/HBV or HIV/HCV coinfections compared to those with monoinfection [32].

Several factors can explain the differences between this study and previous studies. First, this study focused on MSM populations in Northwestern China. This group differs significantly from populations in other regions in terms of socioeconomic status, health behaviors, and access to healthcare resources. Socio-cultural contexts play a critical role in disease management and intervention outcomes. Second, advancements in medical technology, particularly the introduction of direct-acting antivirals (DAAs), have significantly improved the prognosis of HCV-related liver diseases. These advancements were not reflected in some earlier studies. Third, variations in the definitions and assessment criteria for hepatotoxicity across studies may contribute to discrepancies. For instance, whether studies used the U.S. NCI hepatotoxicity grading criteria or other evaluation methods could affect the comparability of results.

This study observed that lower hemoglobin levels appeared to have a protective effect against hepatotoxicity. However, most existing research has focused on the negative associations between anemia and the overall health of HIV-infected individuals [33,34], and no relevant studies were identified linking hemoglobin levels to the occurrence of hepatotoxicity during ART. A significant protective association was observed between lower hemoglobin levels and hepatotoxicity, an unexpected finding that warrants further exploration. One possible explanation is that reduced hemoglobin may reflect a hypoxic state, which activates the HIF pathway and subsequently downregulates major hepatic cytochrome P450 enzymes, leading to decreased drug metabolic activation and attenuated hepatocellular injury [35,36]. While this hypothesis provides a plausible explanation, further studies are needed to confirm the mechanism and clarify its clinical implications

In conclusion, this study represents the first large-scale investigation into ART-related hepatotoxicity among MSM populations in Northwestern China, addressing a critical research gap in the region and uncovering the significant impact of coinfections on ART-related hepatotoxicity. Using Kaplan-Meier curves and Cox regression models, the study conducted a detailed analysis of factors associated with hepatotoxicity, including both any-grade and grade ≥3 hepatotoxicity, providing valuable evidence for clinical stratified management. Through Cox proportional hazards modeling, the study not only confirmed the influence of HBV/HCV coinfections on hepatotoxicity but also, for the first time in Chinese MSM populations, identified older age at ART initiation, a shorter diagnosis-to-treatment interval, and HBV/HCV coinfection as independent risk factors for hepatotoxicity. From a clinical perspective, our findings highlight the importance of dynamic liver function monitoring in HIV-infected MSM with HBV/HCV coinfections during ART. According to the latest U.S. Department of Health and Human Services (DHHS) Guidelines for the Use of Antiretroviral Agents in Adults and Adolescents with HIV (2024),

baseline liver enzyme testing is recommended prior to ART initiation, followed by repeat testing at 2–8 weeks after initiation or regimen change, and every 3–6 months thereafter, with more frequent assessments when using potentially hepatotoxic agents or in high-risk patients [37]. Given the higher hepatotoxicity risk observed in our cohort, we suggest that clinicians consider a more intensive monitoring approach in coinfected patients, particularly during the first year of ART. For example, liver function tests could be conducted every 2–4 weeks initially, with subsequent extension of the interval to 3 months once liver enzymes stabilize. Such an approach may help identify early liver injury, minimize treatment interruptions, and improve the long-term safety of ART in this vulnerable population.

Despite its significant scientific value, this study has several limitations. First, the data were derived from a single medical center, which may limit the generalizability of the findings. Future research involving large-scale, multicenter cohort studies is needed to validate these conclusions. Second, although the study provides valuable insights into hepatotoxicity risk factors, information on alcohol consumption and other potential hepatotoxic exposures (such as use of hepatotoxic medications) was not systematically collected in the medical records. These factors could potentially influence liver outcomes and should be considered in future studies to understand the overall risk profile for hepatotoxicity better. Third, the study did not perform a detailed analysis of the specific impacts of different ART drugs or combination regimens on hepatotoxicity. Future studies should explore the differential effects of various drugs on liver health. Fourth, as an observational study, it is inherently challenging to infer causal relationships. Finally, only six cases of HIV, HBV, and HCV triple infections were identified in this cohort. Due to the very limited sample size, statistical power was insufficient, and the resulting wide confidence intervals could compromise the accuracy of the findings; therefore, these patients were excluded from the main analysis. Furthermore, due to the limited sample size, the study conclusions may not be generalizable to populations with triple infections, and future studies with larger sample sizes are needed to clarify this issue. But previous research has indicated that the incidence of hepatotoxicity is significantly higher in triple-infected individuals compared to those with monoinfection or dual infection, highlighting the need for further investigation into this specific subgroup in future studies.

## Conclusions

In conclusion, this study revealed the epidemiological characteristics of HIV coinfections with HBV and HCV among MSM populations in Northwestern China and their impact on ART-related hepatotoxicity, providing valuable data to optimize regional prevention and treatment strategies. The findings emphasize the importance of early screening, timely ART initiation, and proactive liver function monitoring to mitigate adverse outcomes. By addressing a critical gap in regional data, this research provides valuable evidence to guide personalized treatment strategies and improve public health interventions for managing HIV coinfections.

## Acknowledgments

The authors extend their thanks to the Eighth Hospital of Xi'an, Shaanxi, and all staff who contributed to the data collection and management.

## Author contributions

**Conceptualization:** Xinyu Ma, Ruimin Bai, Yan Zheng.

**Data curation:** Xinyu Ma, Zirong Zhu, Yan Sun.

**Formal analysis:** Xinyu Ma, Tongquan Wang.

**Investigation:** Zirong Zhu, Yan Sun.

**Methodology:** Xinyu Ma, Ruimin Bai, Xinrong Zhao, Yan Zheng.

**Project administration:** Xinyu Ma, Yan Zheng.

**Resources:** Xinyu Ma, Zirong Zhu.

**Software:** Xinyu Ma, Ruimin Bai.

**Supervision:** Ruimin Bai, Xinrong Zhao, Yan Zheng.

**Visualization:** Tongquan Wang.

**Writing – original draft:** Xinyu Ma, Tongquan Wang.

**Writing – review & editing:** Ruimin Bai, Xinrong Zhao, Yan Zheng.

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
