## [Decision Letter · Decision Letter 0]

7 Jul 2025

Dear Dr. Ma,

**ACADEMIC EDITOR: **

clearly state and justify what type of incidence (cumulative incidence or incidence rate) was measuredclearly define hepatotoxicity in the methods section of the abstractto revise the abstract's results section so that the main findings are aligned with the study objectivesthe key words to include the setting/place in which the study was conducted

We look forward to receiving your revised manuscript.

Kind regards,

Cavin Epie Bekolo, MD, MSc

Academic Editor

PLOS ONE

Journal Requirements:

Reviewers' comments:

Reviewer's Responses to Questions

**Comments to the Author**

1. Is the manuscript technically sound, and do the data support the conclusions?

Reviewer #1: Yes

Reviewer #2: Yes

2. Has the statistical analysis been performed appropriately and rigorously?

Reviewer #1: Yes

Reviewer #2: Yes

3. Have the authors made all data underlying the findings in their manuscript fully available?

Reviewer #1: Yes

Reviewer #2: Yes

4. Is the manuscript presented in an intelligible fashion and written in standard English?

Reviewer #1: Yes

Reviewer #2: Yes

Reviewer #1: Dear Dr. Xinyu Ma and colleagues,

Your manuscript “Hepatotoxicity risk factors in HIV-infected MSM with HBV/HCV coinfections: a cohort study in northwest China” offers important insights into the prevalence and impact of HBV and HCV coinfections on ART-related hepatotoxicity in a large MSM cohort. The study is methodologically sound with comprehensive data collection and appropriate analyses. After careful review, I suggest addressing the following minor issues to improve clarity and completeness:

Abstract and Title

1. The title and abstract accurately reflect the scope and main findings. However, consider briefly specifying the retrospective cohort design and inclusion criteria in the abstract methods section for clarity. Expanding slightly on the clinical relevance of the findings in the conclusion would strengthen the abstract.

Introduction

2. The introduction provides a solid background on HIV, HBV, and HCV epidemiology and coinfection risks in MSM. To further contextualize the study, a concise statement discussing gaps in hepatotoxicity data specifically among Chinese MSM on ART would enhance the rationale.

Materials and Methods

3. The retrospective cohort design and inclusion criteria are adequately described. Clarify details on data completeness and handling of missing data, especially for follow-up laboratory results. Also, please specify whether alcohol use or other hepatotoxic exposures were assessed or controlled for, as these could confound outcomes.

Results

4. Results are clearly presented with appropriate use of Kaplan-Meier and Cox regression analyses. The differentiation of hepatotoxicity risks by coinfection type and severity is valuable. Including a table summarizing multivariate hazard ratios for major predictors would improve accessibility.

Discussion

5. The discussion thoughtfully interprets findings in light of existing literature, emphasizing differential hepatotoxicity risks and potential mechanisms. Expanding the commentary on the unexpected protective association of lower hemoglobin and hepatotoxicity with possible explanations would add depth.

Conclusion

6. Conclusions are well supported by data emphasizing the importance of early HBV/HCV screening and liver function monitoring in MSM initiating ART.

Figures and Tables

7. Figures and tables are well formatted and informative. Ensure axis labels and legends are fully explanatory for international readers.

Overall Concerns

8. No scientific or ethical concerns identified. The study shows strong adherence to ethical standards with anonymity and approved protocols.

Competing Interests

9. Disclosures are complete and present no conflicts impacting study validity.

English Language Editing

10. The manuscript is written in clear, professional English. Minor stylistic polishing could enhance readability without substantive changes.

Thank you for your important contribution. I look forward to your revised manuscript addressing these minor comments.

Best regards,

I Ketut Agus Somia

Reviewer

Reviewer #2: This manuscript assesses hepatotoxicity risk factors in HIV-infected MSM with HBV/HCV coinfections in northwest China. The study is thorough, employs suitable statistical methods, and the results support the conclusions. The findings provide valuable evidence for clinicians to optimise ART management in coinfected patients.

Although comprehensive, the study does not fully consider factors such as alcohol consumption or medication adherence, which could influence liver outcomes. Analysis for triple infections (e.g., HIV/HBV/HCV coinfection) was excluded.

Authors should consider elaborating on the limitations, particularly concerning unmeasured confounders such as alcohol consumption, use of traditional medicine, or other hepatotoxic exposures that might influence liver outcomes. Some longer sentences might be better divided for clarity, especially in the Discussion section. Could the authors consider incorporating visual summaries, such as a flowchart or simplified diagram of risk factors? I recommend providing more practical guidance for clinicians, such as recommending the frequency of liver function tests or screening intervals, to make the findings more actionable.

**Do you want your identity to be public for this peer review?** For information about this choice, including consent withdrawal, please see our Privacy Policy

Reviewer #1: **Yes: ** I Ketut Agus Somia

Reviewer #2: **Yes: ** Lawrence Annison

---

## [Author Response · Author response to Decision Letter 1]

21 Aug 2025

Response to Editor’s Comment:

1.clearly state and justify what type of incidence (cumulative incidence or incidence rate) was measured.

Response:

Thank you for your thoughtful comment. We measured hepatotoxicity as an incidence rate rather than cumulative incidence. This decision was based on the fact that our study involved varying follow-up times for participants and included patients who were lost to follow-up. Specifically, the hepatotoxicity incidence rate was calculated as the number of new hepatotoxicity events per 1,000 person-years, which accurately reflects the risk over time for each participant, regardless of differences in follow-up duration. This method is particularly appropriate when dealing with censored data, as it accounts for both event occurrence and the time at risk for each individual. Thus, the measure used in our study is the incidence rate, as indicated by our calculation of new events per person-time.

Specifically, we have clarified this in the Methods section. The revised text now reads:

"Hepatotoxicity incidence and mortality rates were calculated as the number of new hepatotoxicity events and deaths per 1,000 person-years (py) of follow-up. For the incidence rate, hepatotoxicity events were counted annually, and the rate was determined by dividing the number of events by the total person-years at risk, then multiplying by 1,000."

2.clearly define hepatotoxicity in the methods section of the abstract

Response:

Thank you for your insightful comment. In response to your suggestion, we have added a brief definition of hepatotoxicity in the Methods section of the Abstract. While a detailed definition of hepatotoxicity, including the grading system based on the National Cancer Institute (NCI) Common Terminology Criteria for Adverse Events (CTCAE), is provided in the main manuscript, we have now included a concise description in the Abstract for clarity.

The revision in the Abstract now reads as follows:

" Hepatotoxicity was classified into grades 0 (normal) to 4 (life-threatening) according to the degree of elevation in liver enzymes (AST or ALT) ."

This revision ensures that the key concept of hepatotoxicity is clearly stated in the Abstract while maintaining the detailed explanation in the Methods section of the main manuscript.

3.to revise the abstract's results section so that the main findings are aligned with the study objectives

Response:

Thank you for your helpful feedback. In response to your suggestion, we have revised the Results section of the Abstract to ensure it is closely aligned with the study objectives. The revised version emphasizes the prevalence of HBV and HCV coinfections, their significant impact on hepatotoxicity risk, particularly grade ≥3 events, and the clinical factors identified as risk or protective determinants.

The revised Results section now reads as follows:

"Among 4,690 HIV-infected MSM, the prevalence of HIV/HBV and HIV/HCV coinfections was 5.18% and 2.17%, respectively. Coinfected individuals had significantly elevated hepatotoxicity risks. HIV/HBV coinfection substantially increased the risk of any-grade hepatotoxicity (Hazard Ratio [HR]: 1.190, 95% Confidence Interval [CI]: 1.029–1.375) and grade ≥3 hepatotoxicity (HR: 3.161, 95% CI: 2.095–4.769). HIV/HCV coinfection was also associated with a higher risk of any-grade hepatotoxicity (HR: 1.311, 95% CI: 1.050–1.636). Older age at ART initiation, a shorter diagnosis-to-treatment interval, and HBV/HCV coinfection were identified as risk factors, while higher CD4+ T lymphocyte counts and lower hemoglobin levels were protective factors."

4.the key words to include the setting/place in which the study was conducted

Response:

Thank you for your helpful suggestion. In response to your request, we have updated the Keywords section of the manuscript to include "Northwestern China", which reflects the study's geographic setting. The revised Keywords now read as follows:

"HIV, HBV, HCV, MSM, antiretroviral therapy, hepatotoxicity, risk factors, cohort study, Northwestern China"

Response to Reviewesrs' comments:

Reviewer #1:

Abstract and Title

1.The title and abstract accurately reflect the scope and main findings. However, consider briefly specifying the retrospective cohort design and inclusion criteria in the abstract methods section for clarity. Expanding slightly on the clinical relevance of the findings in the conclusion would strengthen the abstract.

Response:

Thank you for your constructive feedback. In response to your suggestion, we have revised the Methods section of the Abstract to explicitly state the retrospective cohort design and to highlight the inclusion criteria for clarity.

The revised Methods section now reads:

“This retrospective cohort study analyzed MSM who were newly diagnosed with HIV and initiated ART in Northwestern China between January 1st, 2005, and June 30th, 2019. A total of 4,690 MSM aged ≥18 years were included and categorized into three groups based on HBV or HCV coinfection status: HIV monoinfection, HIV/HBV coinfection, and HIV/HCV coinfection. Hepatotoxicity was classified into grades 0 (normal) to 4 (life-threatening) according to the degree of elevation in liver enzymes (AST or ALT) . Kaplan-Meier curves were used to evaluate the incidence of hepatotoxicity, mortality, and ART failure rates, while Cox proportional hazards models assessed the independent impact of coinfections on hepatotoxicity risk.”

In response to your comment, we have revised the Conclusion section of the abstract to further emphasize the clinical relevance of our findings. Specifically, we have highlighted the increased risk of hepatotoxicity in HIV-infected MSM with HBV and HCV coinfections and emphasized the importance of early screening, regular liver function monitoring, and standardized antiviral therapy in HIV care. The revised Conclusion now reads as follows:

"HIV/HBV and HIV/HCV coinfections significantly increased the risk of hepatotoxicity in HIV-infected MSM receiving ART. These findings underscore the importance of vigilant liver function monitoring in coinfected patients on ART, particularly in consideration of baseline factors such as age, time to treatment, CD4+ T-cell count, and hemoglobin level, to minimize interruptions and optimize outcomes."

Introduction

2.The introduction provides a solid background on HIV, HBV, and HCV epidemiology and coinfection risks in MSM. To further contextualize the study, a concise statement discussing gaps in hepatotoxicity data specifically among Chinese MSM on ART would enhance the rationale.

Response:

Thank you for your valuable suggestion. In response to your comment, we have revised the Introduction to briefly highlight the paucity of data on ART-related hepatotoxicity in the MSM population in China, particularly in relation to HBV/HCV coinfection. This addition strengthens the rationale and underscores the significance of our study in addressing this gap. The revised text now reads as follows:

“While ART has substantially improved patient outcomes, hepatotoxicity remains a major concern. This risk is particularly pronounced in individuals coinfected with HBV or HCV, whose prevalence continues to rise among MSM in China. However, few studies have systematically investigated ART-related hepatotoxicity and its risk factors in this key population, and data from Northwestern China are especially scarce.”

Materials and Methods

3.The retrospective cohort design and inclusion criteria are adequately described. Clarify details on data completeness and handling of missing data, especially for follow-up laboratory results. Also, please specify whether alcohol use or other hepatotoxic exposures were assessed or controlled for, as these could confound outcomes.

Response:

Thank you for your valuable suggestion. In response to your comment, we have added details regarding data completeness, handling of missing data, and potential hepatotoxic exposures.

Data completeness and missing data:

We clarified that cases with missing baseline or follow-up laboratory results were excluded from the analysis. For other variables with missing values, no imputation was performed; analyses were conducted using available case data.

Potential hepatotoxic exposures:

We also specified that information on alcohol use and other hepatotoxic exposures (e.g., hepatotoxic medications) was not systematically recorded in the medical records, and therefore could not be adjusted for in our analyses. This limitation has been acknowledged in the revised manuscript.

The revisions have been incorporated into the Data sources and collection and Statistical analysis sections of the Materials and Methods. The revised text reads:

Data sources and collection:

“Data completeness was assessed for all variables, particularly follow-up laboratory results (AST and ALT). In cases where baseline laboratory results or follow-up laboratory data were missing, these cases were excluded from the analysis. For any missing values in variables, no imputation was performed, and the analysis was conducted using available case data.”

Statistical analysis:

“Information on alcohol consumption and other potential hepatotoxic exposures (such as the use of hepatotoxic medications) was not systematically collected in the medical records; therefore, these factors were not controlled for in the analysis, which is a limitation of the study.”

Results

4.Results are clearly presented with appropriate use of Kaplan-Meier and Cox regression analyses. The differentiation of hepatotoxicity risks by coinfection type and severity is valuable. Including a table summarizing multivariate hazard ratios for major predictors would improve accessibility.

Response:

Thank you for your helpful suggestion. In response to your comment, we have added the following sentence to the Results section to explicitly reference Table 4, which already provides the requested information regarding the multivariate hazard ratios (HR) and 95% confidence intervals (CI) for the major predictors of hepatotoxicity:

“Table 4 summarizes the multivariate HR for the major predictors of hepatotoxicity and their 95% CI.”

This addition clarifies that Table 4 already includes the necessary information, and we believe it enhances the accessibility and clarity of the results.

Discussion

5.The discussion thoughtfully interprets findings in light of existing literature, emphasizing differential hepatotoxicity risks and potential mechanisms. Expanding the commentary on the unexpected protective association of lower hemoglobin and hepatotoxicity with possible explanations would add depth.

Response:

Thank you for your thoughtful suggestion. In response, we have expanded the Discussion to provide a more detailed and evidence-based interpretation of the unexpected protective association between lower hemoglobin levels and hepatotoxicity. We have introduced several potential mechanisms, including the effects of reduced oxygenation, systemic inflammation, and altered pharmacokinetics, to explain this association. In addition, we have deleted the original, more ambiguous explanation and replaced it with a more comprehensive discussion. The revised discussion now reads as follows:

“A significant protective association was observed between lower hemoglobin levels and hepatotoxicity, an unexpected finding that warrants further exploration. One possible explanation is that reduced hemoglobin may reflect a hypoxic state, which activates the HIF pathway and subsequently downregulates major hepatic cytochrome P450 enzymes, leading to decreased drug metabolic activation and attenuated hepatocellular injury. While this hypothesis provides a plausible explanation, further studies are needed to confirm the mechanism and clarify its clinical implications.

Conclusion

6.Conclusions are well supported by data emphasizing the importance of early HBV/HCV screening and liver function monitoring in MSM initiating ART.

We sincerely thank the Editor for the positive comment. We are pleased that you found our conclusions to be well supported by the data, particularly regarding the importance of early HBV/HCV screening and liver function monitoring in MSM initiating ART.

Figures and Tables

7.Figures and tables are well formatted and informative. Ensure axis labels and legends are fully explanatory for international readers.

Response:

We sincerely appreciate your valuable suggestion regarding the clarity of axis labels and legends for international readers. In accordance with your advice, we have carefully reviewed all figures and tables to ensure that abbreviations are either avoided or explicitly defined.

Specifically, in Table 2, the abbreviation “WBC” has been replaced with “White blood cell counts,” “Platelet” has been replaced with “Platelet counts,” “T.BIL” has been replaced with “Serum total bilirubin,” and “Scr” has been replaced with “Serum creatinine.” In Figure 2, the y-axis label in panel (e) has been revised from “WBC” to “White blood cell counts,” and the label in panel (f) has been changed from “Platelet” to “Platelet counts.” Additionally, definitions for AST (Aspartate aminotransferase) and ALT (Alanine aminotransferase) have been added in the figure legend for Figure 2 to improve clarity.

The modified Figure 2 and its legend are as follows:

Fig 2. Comparison of baseline characteristics of MSM grouped by infection status, including (a) age, (b) age at ART initiation, (c) interval between diagnosis and treatment, (d) CD4+ lymphocyte counts, (e) White blood cell counts, (f) platelet counts, (g) AST, (h) ALT, (i) marital status. Statistically significant pairwise comparison results are indicated as follows: a single asterisk (*) for 0.01 < P < 0.05, double asterisks (**) for 0.001 < P < 0.01, and triple asterisks (***) for P < 0.001. MSM, men who have sex with men; ART, antiretroviral therapy; AST,aspartate aminotransferase; ALT,alanine aminotransferase.

In addition, all abbreviations appearing in the figures and tables of the manuscript have been defined in the figure legends, ensuring consistency with academic writing standards and guidelines.

We believe these revisions enhance clarity and accessibility for an international audience, ensuring that all terms are self-explanatory without requiring prior familiarity with medical abbreviations.

Overall Concerns

8.No scientific or ethical concerns identified. The study shows strong adherence to ethical standards with anonymity and approved protocols.

Response:

We sincerely thank the Editor for recognizing the strong adherence of our study to ethical standards, including the use of anonymity and approved protocols. We greatly appreciate this positive evaluation and encouragement.

Competing Interests

9.Disclosures are complete and present no conflicts impacting study validity.

Response:

We sincerely thank the Editor for acknowledging the completeness of our disclosures and for confirming that no conflicts impact the validity of our study. Additionally, we have included a Competing Interests statement at the end of the manuscript, with the following content: “The authors have declared that no competing interests exist.”

English Language Editing

10.The manuscript is written in clear, professional English. Minor stylistic polishing could enhance readability without substantive changes.

Response:

We sincerely thank your positive feedback on the clarity and professionalism of the manuscript’s language. Following your suggestion, we have carried out a comprehensive English language polishing to further improve readability. All modifications have been clearly marked in the submitted tracked‐changes version of the revised manuscript.

Reviewer #2

1.This manuscript assesses hepatotoxicity risk factors in HIV-infected MSM with HBV/HCV coinfections in northwest China. The study is thorough, employs suitable statistical methods, and the results support the conclusions. The findings provide valuable evidence for clinicians to optimise ART management in coinfected patients.

Although comprehensive, the study does not fully consider factors such as alcohol consumption or medication adherence, which could influence liver outcomes. Authors should consider el

---

## [Decision Letter · Decision Letter 1]

15 Sep 2025

Hepatotoxicity risk factors in HIV-infected MSM with HBV/HCV coinfections: a cohort study in Northwestern China

PONE-D-25-02017R1

Dear Dr. Xinyu Ma,

We’re pleased to inform you that your manuscript has been judged scientifically suitable for publication and will be formally accepted for publication once it meets all outstanding technical requirements.

Kind regards,

Cavin Epie Bekolo, MD, MSc, PhD

Academic Editor

PLOS ONE

Additional Editor Comments (optional):

Reviewer #1:

Reviewer #2:

Reviewers' comments:

Reviewer's Responses to Questions

**Comments to the Author**

Reviewer #1: All comments have been addressed

Reviewer #2: All comments have been addressed

2. Is the manuscript technically sound, and do the data support the conclusions?

Reviewer #1: Yes

Reviewer #2: (No Response)

3. Has the statistical analysis been performed appropriately and rigorously?

Reviewer #1: Yes

Reviewer #2: (No Response)

4. Have the authors made all data underlying the findings in their manuscript fully available?

Reviewer #1: Yes

Reviewer #2: (No Response)

5. Is the manuscript presented in an intelligible fashion and written in standard English?

Reviewer #1: Yes

Reviewer #2: (No Response)

Reviewer #1: Commentary to the Authors

Dear Authors,

Thank you for a comprehensive and clinically important study. Your revisions have markedly improved clarity around the incidence definition, endpoint grading, missing data handling, unmeasured confounding, and figure/table legends. Only minor refinements remain, as detailed below.

*Abstract and Title*

The title and abstract appropriately reflect the study’s key elements. The abstract clearly states the retrospective cohort design, inclusion criteria, CTCAE‑based hepatotoxicity definition, and the main adjusted hazard ratios, and it highlights the clinical implications for liver monitoring in coinfected MSM receiving ART. The strengthened conclusion effectively emphasizes clinical urgency.

Introduction

The introduction now provides adequate background and rationale. The added statement about the limited data on ART‑related hepatotoxicity among Chinese MSM—particularly in Northwestern China—improves contextualization and clarifies the research gap.

Materials and Methods

Methods are clearly described and reproducible. The rationale for reporting incidence per 1,000 person‑years—given variable follow‑up and censoring—is appropriate, and person‑time calculations are explained. Missing data handling is transparent: cases with missing baseline or follow‑up AST/ALT were excluded, no imputation was performed, and analyses used available‑case data. You also correctly note that alcohol use and other hepatotoxic exposures were not systematically collected and therefore could not be controlled for; this is properly framed as a limitation.

Minor consistency issue: Table 3 labels rates as “standardized incidence rates” while the Methods refer to incidence rates per 1,000 person‑years. Please harmonize the terminology or briefly define any standardization applied to avoid confusion.

Results

Results are relevant and plausible. Multivariable hazard ratios demonstrate increased risk of any‑grade hepatotoxicity in both HIV/HBV and HIV/HCV coinfections, and a substantially higher risk of grade ≥3 hepatotoxicity in HIV/HBV infection, consistent with the Kaplan–Meier and Cox analyses.

Please include a brief note on potential residual bias and how covariate adjustment in the Cox models mitigates confounding, while continuing to acknowledge remaining unmeasured hepatotoxic exposures (e.g., alcohol, hepatotoxic medications).

Discussion

The Discussion is well aligned with the findings and existing literature. The different risk patterns observed for HIV/HBV versus HIV/HCV coinfections are thoughtfully explored, with plausible mechanisms and practical implications—particularly the recommendation for intensified monitoring during the first year of ART.

The expanded interpretation around lower hemoglobin and a possible HIF–CYP mechanism adds depth and suggests useful directions for future research.

Conclusion

Conclusions are supported by the data and appropriately emphasize early HBV/HCV screening, timely ART initiation, and tailored liver‑function monitoring based on baseline risk to reduce treatment interruptions and improve outcomes.

Figures and Tables

Figures and tables are clear and legible. Ensure legends remain fully explanatory for international readers.

Overall Concerns

No scientific or ethical concerns were identified.

Declared competing interests appear appropriate and do not seem to bias the study.

No major English language edits are necessary.

Best regards,

I Ketut Agus Somia

udayana University

Reviewer #2: (No Response)

**Do you want your identity to be public for this peer review?** For information about this choice, including consent withdrawal, please see our Privacy Policy

Reviewer #1: **Yes: ** I Ketut Agus Somia

Reviewer #2: **Yes: ** LAWRENCE ANNISON

---

## [Editor Report · Acceptance letter]

PONE-D-25-02017R1

PLOS ONE

Dear Dr. Ma,

I'm pleased to inform you that your manuscript has been deemed suitable for publication in PLOS ONE. Congratulations! Your manuscript is now being handed over to our production team.

Kind regards,

on behalf of

Dr. Cavin Epie Bekolo

Academic Editor

PLOS ONE